# A novel post-processing algorithm for Halo Doppler lidars

Ville Vakkari[1,2], Antti J. Manninen[3], Ewan J. O'Connor[1,4], Jan H. Schween[5], Pieter G. van Zyl[2], Eleni Marinou[6,7]

[1]Finnish Meteorological Institute, Helsinki, FI-00101, Finland
[2]Unit for Environmental Sciences and Management, North-West University, Potchefstroom, ZA-2520, South Africa
[3]Institute for Atmospheric and Earth System Research, University of Helsinki, Helsinki, FI-00014, Finland
[4]Department of Meteorology, University of Reading, Reading, UK,
[5]Institute for Geophysics and Meteorology, University of Cologne, Cologne, Germany
[6]IAASARS, National Observatory of Athens, Athens, 15236, Greece
[7]Institute of Atmospheric Physics, German Aerospace Center (DLR), Oberpfaffenhofen, 82234, Germany

*Correspondence to*: Ville Vakkari (ville.vakkari@fmi.fi)

**Abstract.** Commercially available Doppler lidars have now been proven to be efficient tools for studying winds and turbulence in the planetary boundary layer. However, in many cases low signal-to-noise ratio is still a limiting factor for utilising measurements by these devices. Here, we present a novel post-processing algorithm for Halo Streamline Doppler lidars, which enables an improvement in sensitivity of a factor of five or more. This algorithm is based on improving the accuracy of the instrumental noise floor and it enables using longer integration times or averaging of high temporal resolution data to obtain signals down to -32dB. While this algorithm does not affect the measured radial velocity, it improves the accuracy of radial velocity uncertainty estimates and consequently the accuracy of retrieved turbulent properties. Field measurements with three different Halo Doppler lidars deployed in Finland, Greece and South Africa demonstrate how the new post-processing algorithm increases data availability for turbulent retrievals in the planetary boundary layer, improves detection of high-altitude cirrus clouds, and enables the observation of elevated aerosol layers.

## 1 Introduction

Turbulent mixing in the planetary boundary layer (PBL) is one of the most important processes for air quality, weather and climate (e.g. Garratt, 1994; Baklanov et al., 2011; Ryan, 2016). Mixing layer height (MLH), i.e. the height of the layer that is connected with the surface on timescales of less than 1 hour, is a central parameter describing PBL turbulence (e.g. Seibert et al., 2000). Continuous measurement of MLH with good temporal resolution is not trivial, though. For instance, aerosol backscatter profiles have been commonly used to estimate MLH (Seibert et al., 2000, Pal et al., 2013). The benefit being that aerosol backscatter profiles can be obtained routinely with high temporal resolution (e.g. Emeis et al., 2008), but as this method is not a direct measure of turbulent mixing it is prone to erroneous interpretation especially during morning and evening transition periods of convective PBL (Schween et al., 2014).

Development of fibre-optic Doppler lidar systems during the last 5 to 10 years has enabled direct, long-term observation of MLH with temporal resolutions of typically a few minutes or better (e.g. Tucker et al., 2009; O'Connor et al., 2010; Pearson et al., 2010; Schween et al., 2014; Vakkari et al., 2015; Smalikho and Banakh 2017; Bonin et al., 2017, 2018). Long-range Doppler lidar systems typically have a blind range with a minimum usable distance of 50-100 m, hence scanning Doppler lidar is the only realistic option for covering the full range of MLH from close to ground level up to a few kilometres with good temporal resolution (Vakkari et al., 2015).

In addition to MLH, fibre-optic Doppler lidar systems have enabled also long-term monitoring of horizontal wind profiles within PBL (Hirsikko et al., 2014; Päschke et al., 2015; Newsom et al., 2017; Marke et al., 2018). Together with vertical profiles of higher moments of the velocity distribution (Lothon et al, 2009),  e.g vertical wind speed variance and skewness, as well as turbulent kinetic energy dissipation rate, Doppler lidar measurements enable the diagnosis of the sources of turbulence within PBL (Hogan et al, 2009; Harvey et al., 2013; Tuononen et al., 2017; Manninen et al., 2018).

Velocity measurements with fibre-optic Doppler lidar systems operating at 1.5 µm wavelength depend on light scattering from aerosol particles and cloud droplets as these are small enough to behave as tracers of atmospheric motion. In very clean atmospheric environments, the lack of scattering particles becomes a limiting factor for utilising these systems (e.g. Manninen et al., 2016). Development of new, more powerful yet eye-safe Doppler lidar systems has helped to overcome this limitation to a large degree (e.g. Bonin et al., 2018), yet decreasing the instrumental noise level through post-processing of the data allows the utilisation of weaker signals and can lead to major improvements in data coverage (Manninen et al., 2016). The post-processing algorithm by Manninen et al. (2016) has the added benefit of improving the accuracy of the signal-to-noise ratio (SNR), which leads to more accurate uncertainty estimates of the measured radial velocity (Rye and Hardesty, 1993; Pearson et al., 2009). This is especially important for the retrieval of turbulent properties under weak signal conditions, as uncertainty in instrumental noise level propagates into turbulent properties and wind retrievals (O'Connor et al., 2010; Vakkari et al., 2015; Newsom et al., 2017). Naturally, post-processing methods can be applied to historical data sets as well.

Here we present an improved post-processing algorithm for Halo Photonics Streamline Doppler lidars, which are currently widely-used for PBL research (O'Connor et al., 2010; Pearson et al., 2010; Harvey et al., 2013; Hirsikko et al., 2014; Schween et al., 2014; Päschke et al., 2015; Vakkari et al., 2015; Banakh and Smalikho 2016; Tuononen et al., 2017; Bonin et al., 2018). Building on the work by Manninen et al. (2016), we show that, by changing the way instrumental noise level is determined during periodic background checks, the sensitivity can be improved by as much as a factor of 5; by averaging high time resolution data, signal with an SNR as low as -32dB can be utilised. Case studies from different environments in Finland, Greece and South Africa are presented to demonstrate how the new post-processing algorithm increases data availability for turbulent retrievals in PBL, improves detection of high-altitude cirrus clouds and enables observation of elevated aerosol layers at 2 to 4 km above ground level.

Next, in Section 2 we introduce the Halo Photonics Streamline, Streamline Pro and Streamline XR lidars used in this study. Section 3 describes the improved SNR post-processing algorithm and in Section 4 the three case studies are presented, followed by concluding remarks.

## 2 Instrumentation and measurements

In this study we utilise data from three different versions of Halo Photonics scanning Doppler lidars (Pearson et al., 2009): lidar 46 is a Stream Line system, lidar 53 is a Stream Line Pro system and lidar 146 is a Stream Line XR system. All Halo Photonics Stream Line versions are 1.5 µm pulsed Doppler lidars with a heterodyne detector that can switch between co- and cross-polar channels (Pearson et al., 2009). The Stream Line and the more powerful Stream Line XR lidars are capable of full hemispheric scanning and the scanning patterns are user-configurable. The Streamline Pro version is designed for

harsher environmental conditions with no exterior moving parts, which limits the scanning to within a cone of 20 degrees from vertical. In this study, however, we utilise only vertically pointing measurements in co-polar mode and thus there is no practical difference between the limited and fully scanning versions.

The minimum range for all instruments is 90 m and standard operating specifications for the different versions are given in Table 1. The telescope focus of the Stream Line and Stream Line Pro lidars is user-configurable between 300 m and infinity,

whereas the Stream Line XR focus cannot be changed. Integration time per ray is user adjustable and can be optimised between high sensitivity (long integration time) and high temporal resolution (short integration time) depending on the environmental conditions and research questions. In the measurements utilised in this study, 7s integration time is used for lidars 46 and 53, while lidar 146 is operated with 10s integration time.

In measurement mode the Halo Doppler lidars provide three parameters along the beam direction: radial Doppler velocity

($v_r$), SNR, and attenuated backscatter ($\beta$), which is calculated from SNR taking into account the telescope focus. As part of post-processing, we calculate the measurement uncertainty in $v_r$ ($\sigma_{v_r}$) from SNR according to O'Connor et al. (2010). As discussed earlier, in calculating turbulent parameters from Doppler lidar observations accurate $\sigma_{v_r}$ is needed to differentiate turbulence from instrumental noise (e.g. O'Connor et al., 2010; Vakkari et al., 2015; Newsom et al., 2017).

We present case studies of Halo Doppler lidar measurements at three different locations with three different instruments.

Lidar 53 was deployed at Finokalia, Crete, Greece (35.34°N, 25.67°E) on 8 July 2014. Lidar 46 was deployed at Welgegund, South Africa (26.57°S, 26.94°E) on 6 September 2016 and lidar 146 was deployed at Helsinki, Finland (60.20° N, 24.96° E) on 1 and 6 May 2018.

Additionally, we utilise collocated Raman lidar measurements at Finokalia. These measurements were carried out with the OCEANET PollyXT multiwavelength Raman/polarization lidar system of the Leibniz Institute for Tropospheric Research

(TROPOS). A detailed description of the instrument and its measurements are provided in Engelmann et al. (2016) and Baars et al. (2016), respectively. In brief, PollyXT operates using a Nd:YAG laser that emits light pulses at 1064 nm with a repetition frequency of 20 Hz. The radiation frequency is doubled and tripled, resulting to a simultaneous emitting of 355,

532 and 1064 nm in the atmosphere. The receiver features 12 channels that enable measurements of elastically (three channels) and Raman scattered light (387 and 607 channels for aerosols, 407 for water vapour) as well as depolarization state of the incoming light (355 and 532 nm) and near-range measurements (two elastic and two aerosol Raman channels). In this study, the measurements at 1064 nm are used. The lidar measurements at Finokalia were collected during the 2014 "CHARacterization of Aerosol mixtures of Dust and Marine origin Experiment" (CHARADMExp) on the northern coast of Crete, Greece.

## 3 Improved background check handling algorithm

### 3.1 Signal-to-noise ratio in Halo Doppler lidars

Halo Doppler lidars measure the noise level during periodic background checks, typically once an hour, where the scanner is set to point to an internal (limited-scan) or external target mounted on the instrument itself (hemispheric scan) so that no atmospheric signal is recorded. The raw signal from the amplifier during the background check ($P_{bkg}$) is saved as a profile in ASCII files ("Background_ddmmyy-HHMMSS.txt") with the range resolution configured for the normal measurement mode. For most Stream Line and Stream Line Pro firmware versions, $P_{bkg}$ is written on one line with a fixed precision of six decimals, but varying field width for each range gate. In Stream Line XR firmware, the $P_{bkg}$ value at each range gate is written on its own line.

In most Stream Line and Stream Line Pro instruments, the profile $P_{bkg}(z)$ is flat (constant with range) or presents a small linear increase with increasing distance $z$ from the lidar (Fig. 1a); $P_{bkg}(z)$ following a second-order polynomial can also occur (Manninen et al., 2016). For Stream Line XR instruments, $P_{bkg}(z)$ can vary between a linear and an inverse exponential shape (Fig. 1a), for which the inverse exponential $P_{bkg}(z)$ can be represented as

$$P_{bkg}(z) = \frac{b_1}{\exp(b_2 \cdot z^{b_3})} , \tag{1}$$

where $b_1$, $b_2$, and $b_3$ are scalars and can be determined from a least-squares fit.

In Stream Line and Stream Line Pro lidars, the magnitude of $P_{bkg}$ increases non-linearly with instrument internal temperature (T) (Fig. 1b). For Stream Line XR lidars, which use a different amplifier, the mean $P_{bkg}$ does not depend on T, however, the amplifier alternates randomly between a high mode ($P_{bkg} \approx 3.6 \ 10^8$ for lidar 146) and a low mode ($P_{bkg} \approx 3.2 \ 10^8$ for lidar 146) as seen in Fig. 1c. Furthermore, it appears that the inverse exponential $P_{bkg}$ shape occurs only in the low mode, but not all low mode $P_{bkg}$ profiles follow Equation 1.

The Halo Doppler lidar firmware accounts for changes in $P_{bkg}$ level by calculating SNR as

$$SNR_0 = \frac{A_0 \cdot P_0(z)}{A_{bkg} \cdot P_{bkg}(z)} - 1 , \tag{2}$$

where $P_0(z)$ is the raw signal from the amplifier during each measurement, $P_{bkg}(z)$ has been obtained during the previous background check, and scalar scaling factors $A_0$ and $A_{bkg}$ are determined on-line for each $P_0$ and $P_{bkg}$ profile. Here, we denote the unprocessed SNR output by the instrument as $SNR_0$. Note that $A_0$ and $A_{bkg}$ are not saved by the firmware, which means that the high and low mode in Stream Line XR lidars cannot be identified in the $SNR_0$ time series.

5    Equation (2) is straightforward to determine on-line as there are no assumptions about the shape of $P_{bkg}$ and it gives a reasonably good first estimate of SNR. However, Equation (2) is vulnerable to inaccuracy in determining $A_0$ and $A_{bkg}$ as well as to any deviation from the actual noise level during measurement of $P_{bkg}$. An offset in $A_0$ inflicts a constant offset in $SNR_0$ in a single profile, while an offset in $A_{bkg}$ does the same for all profiles between two background checks (c.f. Manninen et al., 2016). The magnitude of typical offset in $A_0$ and $A_{bkg}$ varies from instrument to instrument; in some cases they can have a major effect on data coverage (Manninen et al., 2016).

In all Halo Doppler lidars $P_{bkg}$ contains a small but varying offset from the actual noise level at each range gate because of the finite duration of the background check. These offsets appear as a small constant offset in $SNR_0$ at each range gate between two background checks. To minimise the effect of offset in $P_{bkg}$, the integration time of background check measurement was originally designed to be 6 times as long as the integration time in measurement mode. The duration of the background check is user-configurable, however, for long integration times of up to 6 minutes considered in this manuscript such long background checks are not a viable option. In the next section we present an improved algorithm to correct $SNR_0$ for inaccuracies in $A_0$, $A_{bkg}$ and $P_{bkg}$.

## 3.2 Improved SNR post-processing algorithm

Whether $P_{bkg}(z)$ is linear or follows some other functional form is readily determined by fitting expected functions to it (c.f. Manninen et al., 2016). For Stream Line and Stream Line Pro lidars we consider second order polynomial to represent $P_{bkg}(z)$ better than a linear fit if it has at least 10 % lower root-mean-squared (RMS) error than the linear fit to $P_{bkg}(z)$. For Stream Line XR we consider Equation (1) to represent $P_{bkg}(z)$ better if it has at least 5 % lower RMS error than the linear fit to $P_{bkg}(z)$. Furthermore, knowing the typical noise level of a certain instrument, a RMS threshold can be applied to discard bad fits and to flag periods of increased uncertainty.

25    Denoting the selected fit to $P_{bkg}(z)$ as $P_{fit}(z)$, the residual is

$$P_{bkg,res}(z) = P_{bkg}(z) - P_{fit}(z) . \qquad (3)$$

Averaging $P_{bkg,res}(z)$ over a large number of $P_{bkg}(z)$ profiles reveals a persistent structure in the residual (Fig. 2a). This part of $P_{bkg,res}(z)$ originates in the amplifier response to the transmitted pulse, denoted here as $P_{amp}(z)$, and it is the main reason for using the gate-by-gate defined $P_{bkg}(z)$ profile in SNR calculation by the manufacturer. However, $P_{amp}(z)$ stays reasonably constant over time and can be obtained from a long enough data set of $P_{bkg}(z)$. Here, we used a discrete wavelet transform with Symmlet order 8 wavelet as a low-pass filter to de-noise the averaged $P_{bkg,res}(z)$. As shown in Fig. 2a, $P_{amp}(z)$ is instrument-specific and needs to be determined individually for each device.

In Stream Line and Stream Line Pro lidars, T has a small effect on $P_{amp}(z)$ as seen in Fig.2b. However, this can be addressed based on a suitably long T data set and $P_{bkg,res}(z)$ by determining $P_{amp}(z)$ as a function of the internal temperature. In practice, at least 300 $P_{bkg}(z)$ profiles are required to obtain a reliable estimate of $P_{amp}(z)$. Consequently, for an 11-month measurement campaign at Welgegund, we could determine $P_{amp}$ as a function of T at 1°C resolution from 25 to 31 °C (Fig. 2b). For T<23°C or T>35°C we could only determine aggregate $P_{amp}$ profiles, but then these temperature ranges comprise only 9 % of the measurements in this data set. For optimal data quality, additional temperature stabilisation could be applied to ensure that $P_{amp}$ is always in the well-characterised temperature range.

In Stream Line XR lidars, $P_{amp}$ does not depend on T, however, $P_{amp}$ has to be determined separately for the high and low mode of $P_{bkg}$ (cf. Fig. 1c). For lidar 146, we define $P_{bkg}$ high mode as mean $P_{bkg} > 3.4\ 10^8$ and $P_{bkg}$ low mode as mean $P_{bkg} < 3.4\ 10^8$, respectively. As seen in Fig. 2c, $P_{amp}$ for these two modes differ substantially.

We consider the sum of $P_{fit}(z)$ and $P_{amp}(z)$ as the best estimate for the actual instrumental noise level during a background check,

$$P_{noise}(z) = P_{fit}(z) + P_{amp}(z) \ . \tag{4}$$

Using Equation (2), we can move from a $P_{bkg}$-based SNR (i.e. $SNR_0$) to a $P_{noise}$-based, corrected SNR (denoted here as $SNR_1$) simply as

$$SNR_1(z) = (SNR_0(z) + 1) \cdot \frac{P_{bkg}(z)}{P_{noise}(z)} - 1 \ , \tag{5}$$

Next, we utilise the Manninen et al. (2016) algorithm to identify any possible bias in the $A_0$ to $A_{bkg}$ ratio. In short, Manninen et al. (2016) cloud and aerosol screening is applied first to time series of $SNR_1$. Note that typically cloud and aerosol signal is easier to discern in $SNR_1$ than in $SNR_0$ and thus cloud screening is applied after Equation 5. Then, first and second order polynomial fits are calculated for each cloud-screened profile of $SNR_1(z)$ and a RMS threshold is used to select the appropriate fit, similar to determining $P_{fit}(z)$. Denoting the selected fit to cloud and aerosol free measurements as $SNR_{fit}(z)$ we obtain

$$SNR_2(z) = \frac{SNR_1(z)+1}{SNR_{fit}(z)+1} - 1 \ , \tag{6}$$

which is our final corrected SNR.

Note that to correct only for the bias in the $A_0$ to $A_{bkg}$ ratio, a scalar denominator in Equation (6) would be sufficient. However, using the fitted profile $SNR_{fit}(z)$ as the denominator accounts for possible changes in the slope of $P_{noise}$ since the last background check.

### 3.2.1 Implications for Stream Line XR lidars

The calculation of $SNR_2$ with Equation (6) relies on the fitting to cloud and aerosol free measurements. For Stream Line and Stream Line Pro lidars, which do not exhibit the inverse exponential $P_{bkg}$ shape, $SNR_{fit}(z)$ will capture the shape of the actual noise level in nearly all cases. However, for Stream Line XR lidars the randomly occurring inverse exponential $P_{bkg}(z)$ shape (Fig. 1a) is almost always masked by aerosol and/or cloud signal during measurement. Thus, it is not possible to correct for changes in shape of $P_{noise}(z)$ with $SNR_{fit}(z)$ during post-processing. However, the magnitude of uncertainty in $P_{noise}(z)$ can be estimated from the average depth of the inverse exponential dip in $P_{bkg}(z)$ during background checks (c.f. Fig. 3a).

Now as $A_0$ is not saved, it is not possible to tell whether the amplifier was operating in high or low mode during measurement. Consequently, the difference in $P_{amp}$ for the amplifier high and low modes also adds to the uncertainty in SNR for Stream Line XR systems, but, compared to the effect of the inverse exponential shape of $P_{bkg}(z)$, the effect of $P_{amp}$ is approximately 10 times smaller. However, any possible bias in the $A_0$ to $A_{bkg}$ ratio can be corrected and this is readily done by applying a linear fit to $SNR_1(z)$ at range gates 100-400 (where SNR is not affected by inverse exponential $P_{bkg}$) and using this as $SNR_{fit}(z)$ in Equation (6).

In practice, there are two options for SNR post-processing for Stream Line XR lidars. The first option is to accept the fitted Equation (1) for $P_{fit}(z)$ when it describes $P_{bkg}(z)$ better. With this approach $SNR_2$ may overestimate the actual SNR if the shape of $P_{noise}$ changes from Equation (1) to linear after the background check. Correspondingly, a change from a linear $P_{noise}$ to the inverse exponential shape during measurement results in $SNR_2$ underestimating the actual SNR.

The second option for Stream Line XR SNR post-processing is to calculate a linear fit to $P_{bkg}(z)$ based on range gates 100-400 and always use this for $P_{fit}(z)$. In this case, we use only high mode $P_{amp}(z)$ in calculating the noise level (Equation 4) and denote it as $P'_{noise}(z)$. Consequently $SNR'_2$ is the lower limit of the actual SNR, which can be useful if SNR-threshold is used to determine usable signal for further analysis. For lidar 146 background checks from 12 January 2018 to 31 May 2018, the underestimation was on average 0.5% of SNR+1 at the first usable range gate and decreased rapidly with increasing range (Fig. 3a). In the worst case, the underestimation at the first usable range gate was 3% of SNR+1.

Uncertainty in SNR leads to uncertainty in $\sigma_{vr}$, as $\sigma_{vr}$ is mostly a function of SNR (Pearson et al., 2009). However, $\sigma_{vr}$ decreases rapidly with increasing SNR (Fig. 3b). Therefore, even the worst-case underestimation in SNR has only a limited effect on $\sigma_{vr}$ if SNR is even moderately high (> 0.03, -15.2 dB). On the other hand, for observations > 2000 m away from the lidar, where signals are typically low, the uncertainty in SNR is also low (Fig. 3a). In the end, uncertainty in SNR and its effects in β and $\sigma_{vr}$ need to be evaluated individually for each profile in Stream Line XR lidars.

## 4 Case studies

### 4.1 Welgegund 6 September 2016

On 6 September 2016 lidar 46 was operating at Welgegund, South Africa and a time series of SNR in vertically-pointing measurement mode for this day is presented in Figure 4. In this case, $SNR_0$ is very close to 0 when there are no clouds or

aerosol present (Fig. 4a), indicating that the on-line calculation of $A_0$ and $A_{bkg}$ is quite successful. However, the presence of small but varying offsets in $P_{bkg}(z)$ is apparent in Fig. 4a as horizontal stripes in $SNR_0$ time series between the background checks conducted on the hour.

In the $SNR_1$ time series (Fig. 4b) the horizontal stripes have been removed by applying the smooth $P_{noise}$-based background with Equation 5. At the same time, the elevated aerosol layer at 2000 – 4000 m above ground level (a.g.l.) becomes easily

discernible. Small biases in the $A_0$ to $A_{bkg}$ ratio become visible as vertical stripes, for instance between 5-6 UTC in Fig. 4b, which are then corrected for in the time series of $SNR_2$ (Fig. 4c).

Comparing the standard deviation of SNR ($\sigma_{SNR}$) for cloud and aerosol free range gates shows a clear improvement in the noise level with the new post-processing algorithm (Fig. 4d). The main advantage of the new post-processing algorithm is that it enables averaging SNR; for $SNR_0$ any offsets in $P_{bkg}(z)$ become the limiting factor. This is clearly seen in Fig. 4d,

where $\sigma_{SNR}$ for $SNR_2$ decreases with increasing integration time per profile following closely $\sigma/\sqrt{N}$ rule as expected, but increasing integration time has little effect on $\sigma_{SNR}$ for $SNR_0$.

Figure 5 demonstrates how the lower noise floor with the new post-processing algorithm allows determining vertical wind speed variance ($\sigma^2_w$) up to 2000 m a.g.l. (i.e. up to the top of the mixed layer) on this day. Furthermore, by averaging the originally 7s data to 168s integration time per profile and applying the new post-processing algorithm, the SNR threshold at

the $3\sigma$-level (c.f. Fig. 4d) can be decreased from 0.0032 (-25dB) for $SNR_0$ to 0.00065 (-32dB) for $SNR_2$. Consequently, $\beta$ can be retrieved for the elevated aerosol layer at 2000 – 4000 m a.g.l. (Fig. 5c,d). Note that the offsets in $P_{bkg}(z)$ result in horizontal stripes in the 168s integration time $\beta$ calculated from $SNR_0$ in Fig. 5c.

Lower noise floor enables also wind retrievals with a lower SNR-threshold, which increases the data availability. The effect on data availability depends on atmospheric conditions, though. In this case for instance (Welgegund, 6 September 2016) a

75° elevation angle velocity azimuth display (VAD) scan was utilised for horizontal wind retrieval every 15 minutes. With the new post-processing algorithm the SNR-threshold for wind retrieval could be decreased from 0.0045 to 0.0032. This decrease in SNR-threshold enabled wind retrieval for 2-13 range gates more from each VAD scan; on average, winds could be determined from 7.5 additional range gates per VAD scan. That is, vertical coverage of wind retrievals increased on average by 200 m with the new post-processing.

Wind retrievals at lower SNR will have higher uncertainty due to higher instrumental noise in radial velocity measurement, yet enhanced SNR will enable more accurate determination of the instrumental uncertainty in wind retrievals. However, as a major fraction of the uncertainty in retrieved winds arises in atmospheric turbulence (Newsom et al., 2017) the more accurate

SNR will have only a limited effect on the overall uncertainty in the wind retrieval. Therefore, the uncertainty in each wind retrieval should be evaluated e.g. with the methodology of Newsom et al. (2017) before the wind retrievals are disseminated.

### 4.2 Helsinki 1 and 6 May 2018

Measurements with lidar 146 at Helsinki, Finland, on 6 May 2018 (Fig. 6) present all the issues with a Stream Line XR lidar
at its worst. In Fig. 6a, $SNR_0$ is negative e.g. from 1 to 5 UTC, 6 to 10 UTC and 12 to 13 UTC because of an erroneous $A_{bkg}$ coefficient. On the other hand, the individual profiles with unrealistically high $SNR_0$ around 11 UTC, 14 to 15 UTC and 20 to 21 UTC indicate errors in the $A_0$ coefficient. Additionally, horizontal stripes in $SNR_0$ time series similar to lidar 46 (Fig. 4a) indicate offsets in $P_{bkg}(z)$. The reason for poor determination of $A_0$ and $A_{bkg}$ for lidar 146 seems to be that $P_{bkg}(z)$ is frequently non-linear, unlike for lidar 46 for example.

The new post-processing algorithm corrects the errors in $A_0$ and $A_{bkg}$ as well as the stripes due to offsets in $P_{bkg}(z)$ as seen in Fig. 6c. However, Fig. 6c shows that $P_{bkg}(z)$ changing between the inverse exponential and linear shape causes over- and underestimation of $SNR_2$ in the lowest 1500 m a.g.l.. For instance, positive $SNR_2$ in the lowest 1000 m at 12 to 13 UTC and negative $SNR_2$ in the lowest 1000 m at 14 to 15 UTC are due to noise level shape changes between background check and measurement modes. During these periods, the lidar signal is fully attenuated by a cloud within the lowest 200 m, and
consequently $SNR_2$ in the 200-1000 m range should be zero. In Fig. 6d, $SNR'_2$ is calculated using a linear fit only to $P_{bkg}(z)$ as discussed in Section 3.2.1. This removes the overestimate of SNR at 12-13 UTC, but cannot correct the underestimates.

Measurements with lidar 146 on 1 May 2018 at Helsinki present much less noisy $SNR_0$ than on 6 May 2018 as seen in Fig. 7a. On this day there are cirrus clouds present at 8000-12000 m a.g.l., but the stripes due to offsets in $P_{bkg}(z)$ make it difficult to distinguish the clouds from noise in $SNR_0$. Applying the new post-processing algorithm and increasing integration time
from 10s to 60s for this day enables the SNR threshold at 3σ-level to be lowered from 0.0035 (-24.5dB) for $SNR_0$ to 0.0012 (-29dB) for $SNR_2$. This results in a significant increase in data coverage for the cirrus clouds as shown in Figs. 7c and 7d.

### 4.3 Finokalia 8 July 2014

Time series of SNR in vertically-pointing measurement with lidar 53 on 8 July 2014 at Finokalia, Greece, is presented in Fig. 8. On this day, $SNR_0$ is close to 0 for 4000-9600 m a.g.l. elevation (Fig. 8a), indicating that the on-line calculation of $A_0$
and $A_{bkg}$ is quite successful. Only at 0-1 UTC and 20-21 UTC is $SNR_0$ negative indicating a small offset in $A_{bkg}$. However, horizontal stripes in the $SNR_0$ time series between the background checks are apparent in Fig. 8a, indicating the presence of small but varying offsets in $P_{bkg}(z)$.

After SNR post-processing (Fig. 8b), elevated aerosol layers at 1000-4000 m a.g.l. are clearly visible on this day. These aerosol layers were also observed with a co-located multi-wavelength Raman lidar Polly XT (Baars et al., 2016; Engelman et
al., 2016). A comparison of lidar 53 and the Raman lidar measurements at 1064 nm wavelength are presented in Fig. 9. Considering the wavelength difference, the agreement between the two systems is reasonably good. Further averaging of $SNR_2$, in this case up to 350s integration time, allows the determination of β for the elevated aerosol layers. With this long

integration time we can reach a 3σ SNR threshold of 0.00059 (-32dB) for $SNR_2$. For $SNR_0$, offsets in $P_{bkg}(z)$ are the limiting factor in determining the SNR threshold and at 3σ level only 0.0044 (-24dB) can be achieved.

**5 Conclusion**

In this paper we have presented an improved SNR post-processing algorithm for Halo Doppler lidars. For Stream Line and
Stream Line Pro lidars this method enables accurate SNR and β retrievals from the first usable gate onwards. For Stream Line XR lidars we identified a previously unknown source of uncertainty in the near-range (<1500 m) SNR due to variations in the noise floor of these systems. We present a method to estimate the magnitude of this source of uncertainty, although it cannot be completely eliminated.

We have shown that defining the noise floor on a point-by-point basis during periodic background checks results in a small,
variable offset in SNR at each range gate. This offset is due to finite duration of the background check and becomes the limiting factor in retrieving weaker signals with Halo Doppler lidars, or with any system based on such a point-by-point defined noise floor. The improved SNR post-processing algorithm removes this source of error by introducing a more accurate, continuous noise floor. Independent of the noise floor, on-line scaling of raw signal from the amplifier by the firmware fails occasionally. This source of error in SNR was targeted by Manninen et al. (2016) and their algorithm is
adapted here as part of the improved SNR post-processing algorithm (Equation 6). Correcting for these two sources of error in SNR enables retrieving data at much lower SNR than before. Increasing integration time per profile to a few minutes, SNR down to $6 \cdot 10^{-5}$ (-32 dB) can be utilised.

Our analysis shows that even if the technical specifications of two Doppler lidar systems are identical, their instrumental noise characteristics can be quite different (Fig. 2). Therefore, the lidar operator should inspect each system individually to
ensure highest data quality. Note that this algorithm or similar processing is needed to define the instrumental noise level even if raw spectra are utilised instead of the processed in data. The algorithm presented here can be applied in semi-operational use as long as at least 300 background checks (acquired in two weeks of measurements with typical configuration) are available for characterising the amplifier response to the transmitted pulse. A Matlab implementation of this algorithm is available through Github (Manninen, 2019).

We have demonstrated that the improved SNR post-processing can help retrieving turbulent properties up to the top of the mixed layer under low aerosol load. With enhanced SNR, the instrumental noise contribution to radial velocity variance can be estimated with better accuracy, which will improve the quality of turbulent parameter retrievals. The reduced noise floor enables horizontal wind retrievals with a lower SNR-threshold and increases data availability, depending on atmospheric conditions. Furthermore, we have demonstrated that a combination of reduced noise floor and increased integration time
allow detection of elevated aerosol layers with Stream Line and Stream Line Pro lidars. Even for the more powerful Stream Line XR lidars, the new SNR post-processing can increase data availability e.g. in case of high altitude cirrus clouds. In conclusion, the improved SNR post-processing introduced in this paper enhances the capabilities of Halo Doppler lidars in

studying atmospheric turbulence in weak signal conditions and opens up new possibilities for studying elevated aerosol layers, such as volcanic ash, Aeolian dust or biomass burning smoke.

**Appendix A:**

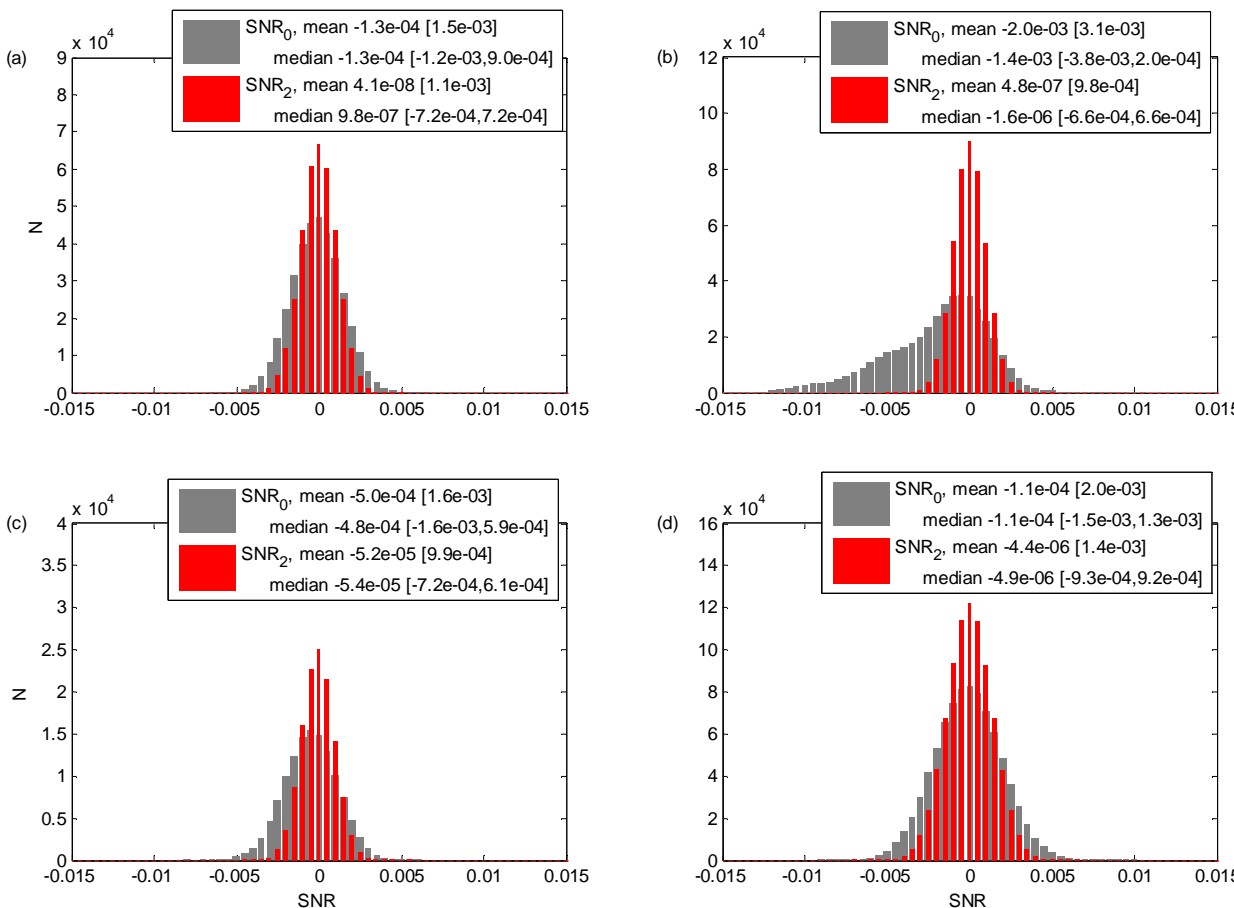

5 **Figure A1. Histograms of $SNR_0$ and $SNR_2$ in cloud and aerosol free regime for the four case studies considered in Section 4. For each case, mean [standard deviation] and median [25th, 75th percentile] of $SNR_0$ and $SNR_2$ are included. (a) Welgegund on 6 September 2016, 00-24 h UTC, 4800-9000 m a.g.l.. (b) Kumpula on 1 May 2018, 02-24 h UTC, 6000-12000 m a.g.l.. (c) Kumpula on 6 May 2018, 00-12 h UTC, 4000-7000 m a.g.l.. (d) Finokalia on 8 July 2014, 00-24 h UTC, 5000-96000 m a.g.l..**

**Acknowledgements**

10 We gratefully acknowledge financial support by TOPROF (COST Action ES1303) and North-West University for measurements at Welgegund. CHARADMExp was an experimental campaign of the National Observatory of Athens (NOA) and have been supported from the ESA-ESTEC project "Characterization of Aerosol mixtures of Dust And Marine origin"

contract no. IPL-PSO/FF/lf/14.489. We acknowledge Ronny Engelmann and Holger Baars from TROPOS for providing Raman lidar data and financial support through the High-Definition Clouds and Precipitation for advancing Climate Prediction research program (HD(CP)2; FKZ: 01LK1209C and 01LK1212C) funded by Federal Ministry of Education and Research in Germany (BMBF), ACTRIS under grant agreement no. 262254 of the European Union Seventh Framework

Programme (FP7/2007-2013) and ACTRIS-2 under grant agreement no. 654109 from the European Union's Horizon 2020 research and innovation programme.

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

**Table 1 Specifications for Halo Doppler lidars utilised in this study.**

| | |
|---|---|
| Lidar number and version | 46, Stream Line |
| | 53, Stream Line Pro |
| | 146, Stream Line XR |
| Wavelength | 1.5 μm |

| | |
|---|---|
| Pulse repetition rate | 15 kHz (46 and 53) or |
| | 10 kHz (146) |
| Nyquist velocity | 20 m s$^{-1}$ |
| Sampling frequency | 50 MHz |
| Velocity resolution | 0.038 m s$^{-1}$ |
| Points per range gate | 10 |
| Range resolution | 30 m |
| Maximum range | 9600 m (46 and 53) or |
| | 12000 m (146) |
| Pulse duration | 0.2 μs |
| Lens diameter | 8 cm |
| Lens divergence | 33 μrad |
| Telescope | monostatic optic-fibre |
| | coupled |

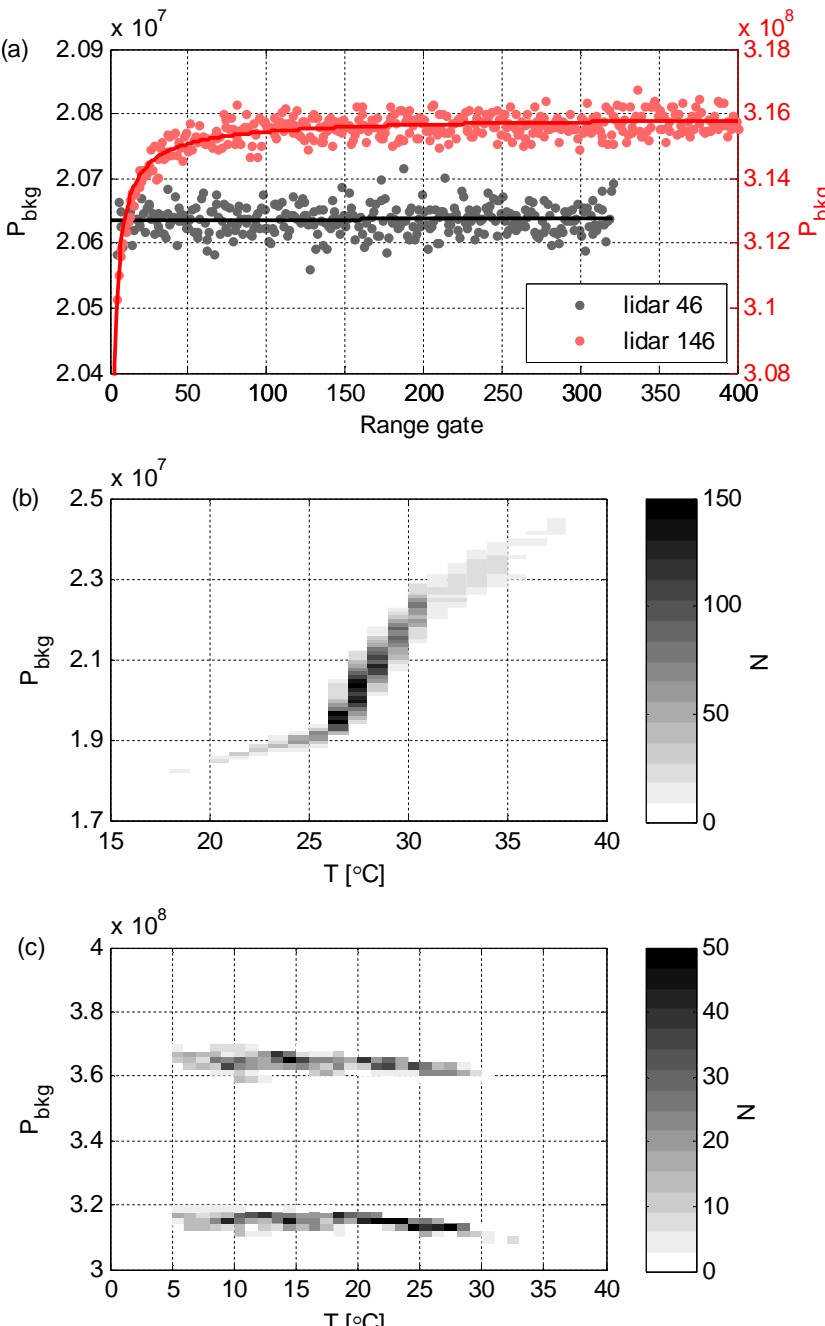

**Figure 1: (a)** $P_{bkg}$ **measured by lidar 46 on 6 September 2016 at 23h UTC and** $P_{bkg}$ **measured by lidar 146 on 1 May 2018 at 10h UTC. Also** $P_{fit}$ **is indicated for both systems. (b) 2D-histogram of mean** $P_{bkg}$ **vs. T for lidar 46. (c) 2D-histogram of mean** $P_{bkg}$ **vs. T for lidar 146.**

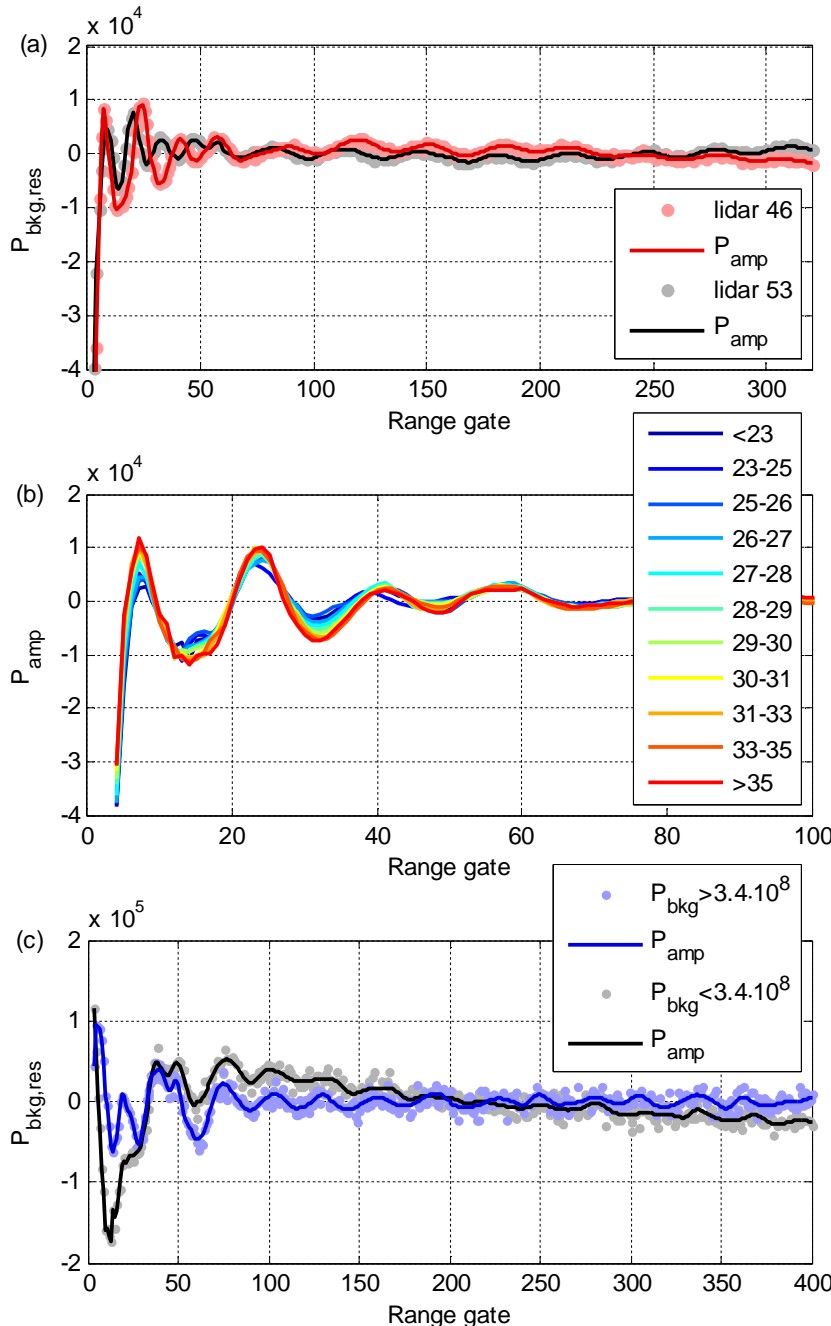

**Figure 2: (a) Lidar 46 $P_{bkg,res}$ averaged from 20 August 2016 to 14 June 2017 (7193 background checks). Lidar 53 $P_{bkg,res}$ averaged from 1 January 2014 to 30 November 2015 (16802 background checks). $P_{amp}$ is plotted for both systems. (b) First 100 range gates of lidar 46 $P_{amp}$ calculated for different ranges of T. (c) Lidar 146 $P_{bkg,res}$ averaged from 12 January 2018 to 31 May 2018. $P_{bkg,res}$ is averaged separately for high $P_{bkg}$ mode (1375 background checks) and for low $P_{bkg}$ mode (1623 background checks). $P_{amp}$ is plotted for both modes.**

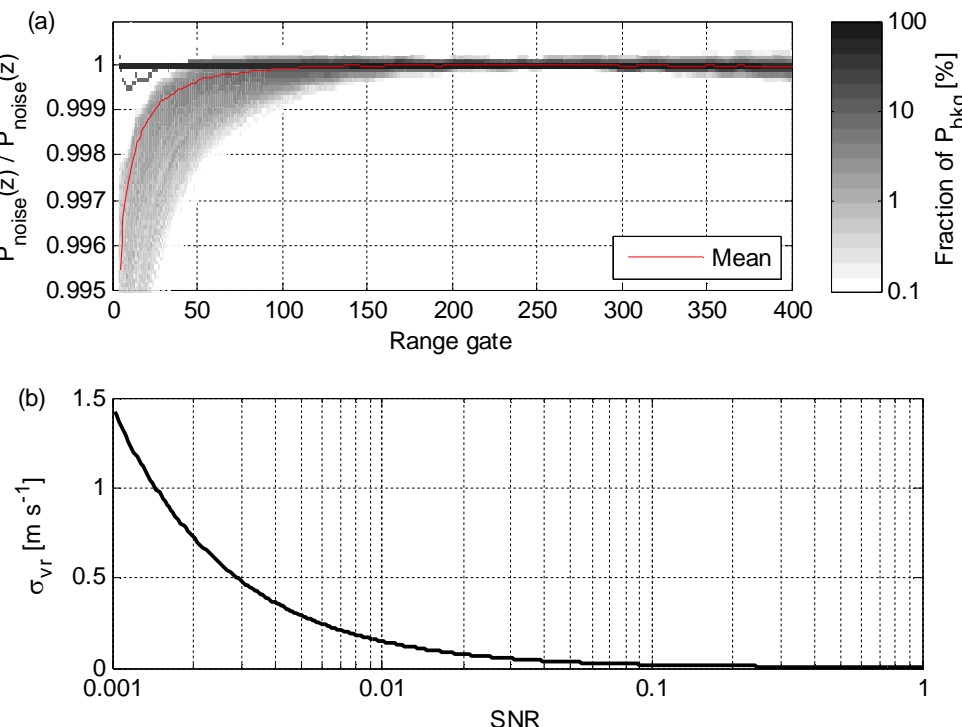

**Figure 3: (a) 2D-histogram of the ratio of $P_{noise}(z)$ (best estimate) to $P'_{noise}(z)$ (linear fit only) for lidar 146 background checks. Mean of the ratio is also indicated. (b) $\sigma_{vr}$ as a function of SNR for lidar 146.**

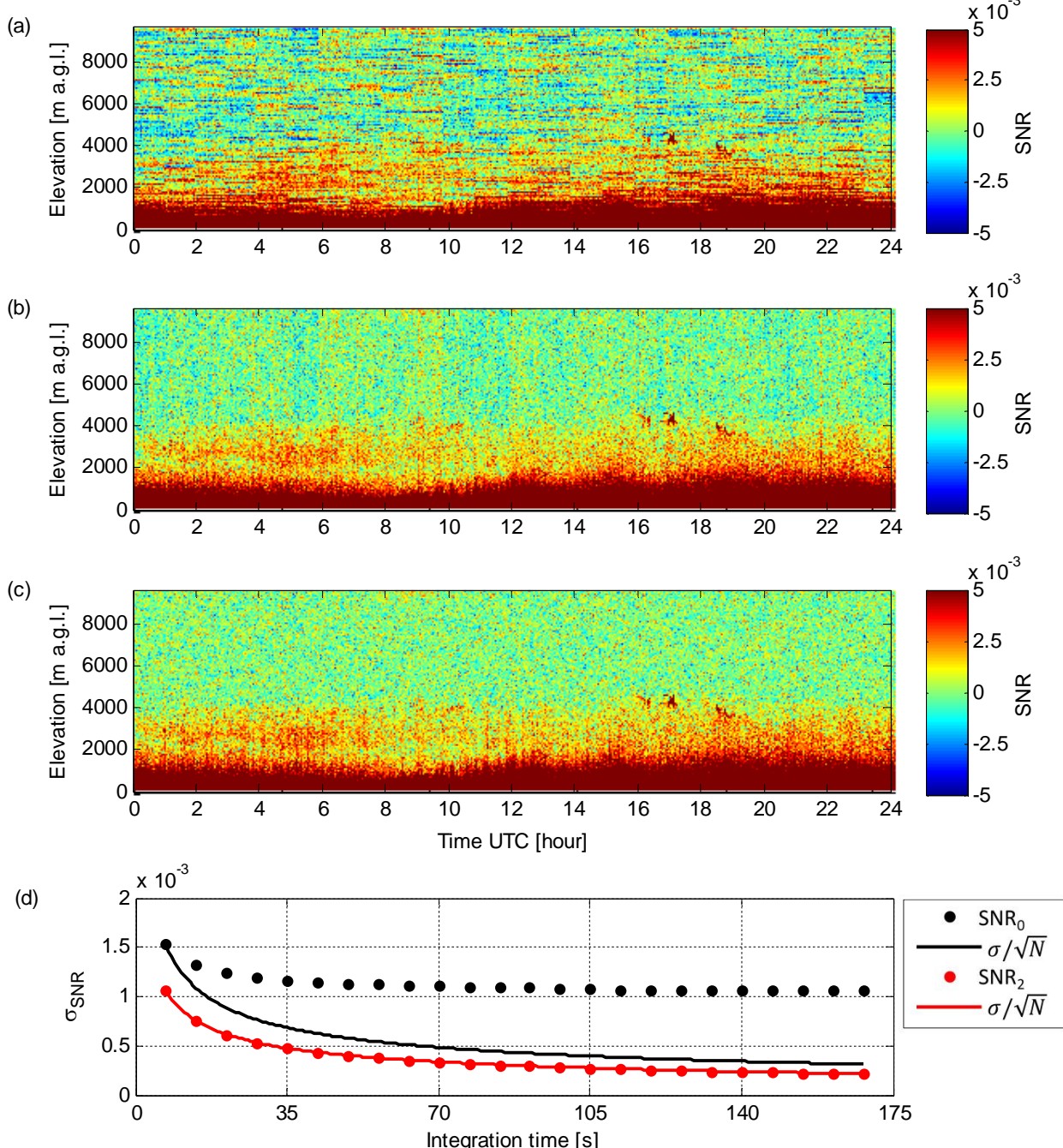

**Figure 4: Data from lidar 46 at Welgegund on 6 September 2016. (a)** Time series of $SNR_0$ profile in vertically-pointing mode. **(b)** Time series of $SNR_1$ profile in vertically-pointing mode. **(c)** Time series of $SNR_2$ profile in vertically-pointing mode. **(d)** $\sigma_{SNR}$ as a function of integration time per profile for $SNR_0$ and $SNR_2$ for range gates at 4800 – 9000 m a.g.l.. Also $\sigma/\sqrt{N}$, where $\sigma$ is $\sigma_{SNR}$ at integration time of 7s (original integration time per profile) and N is the number of averaged profiles, is included in (d).

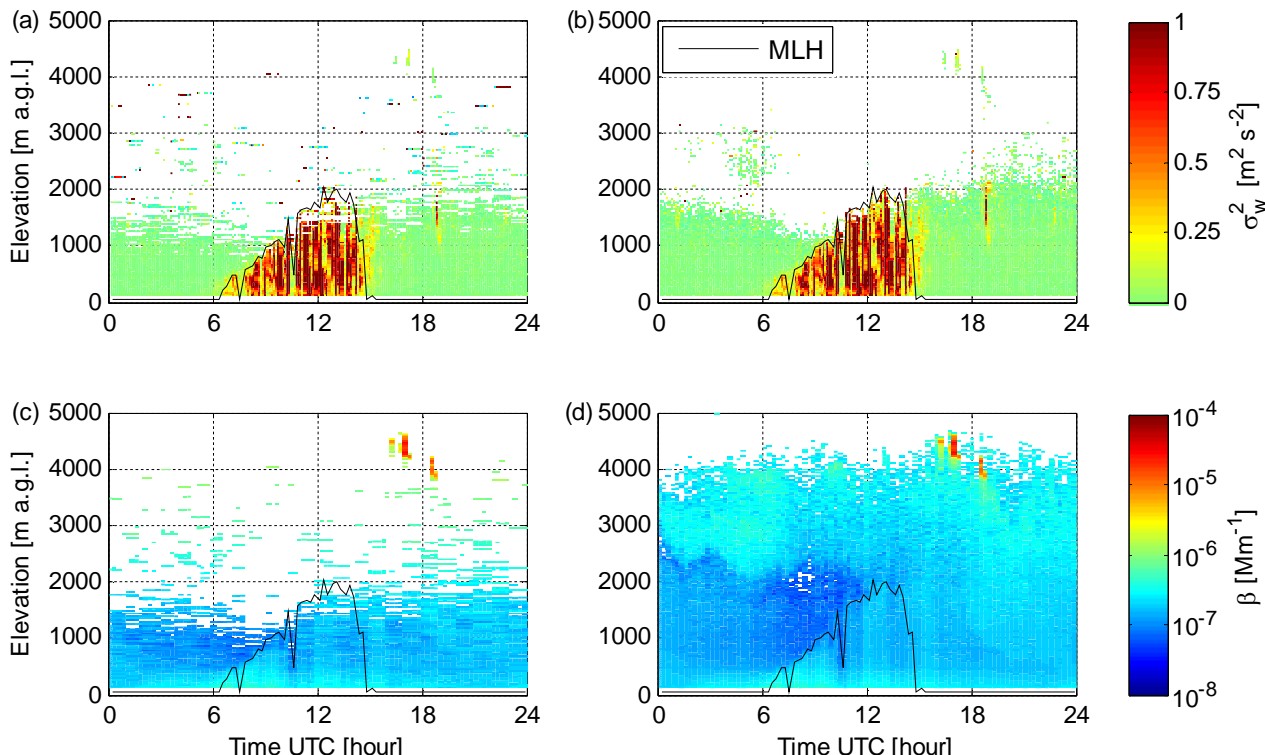

**Figure 5: Data from lidar 46 at Welgegund on 6 September 2016. (a) Time series of $\sigma^2_w$ profile, where a threshold of 0.0031 (2σ) has been applied to $SNR_0$. (b) Time series of $\sigma^2_w$ profile, where a threshold of 0.0021 (2σ) has been applied to $SNR_2$. In (a) and (b) the instrumental noise contribution to $\sigma^2_w$ has been subtracted. (c) Time series of β obtained with 168s integration time from $SNR_0$. β has been filtered with a threshold of 0.0032 (3σ) applied to $SNR_0$. (d) Time series of β obtained with 168s integration time from $SNR_2$. β has been filtered with a threshold of 0.00065 (3σ) applied to $SNR_2$. Mixing layer height (MLH) determined from panels b and d is also indicated.**

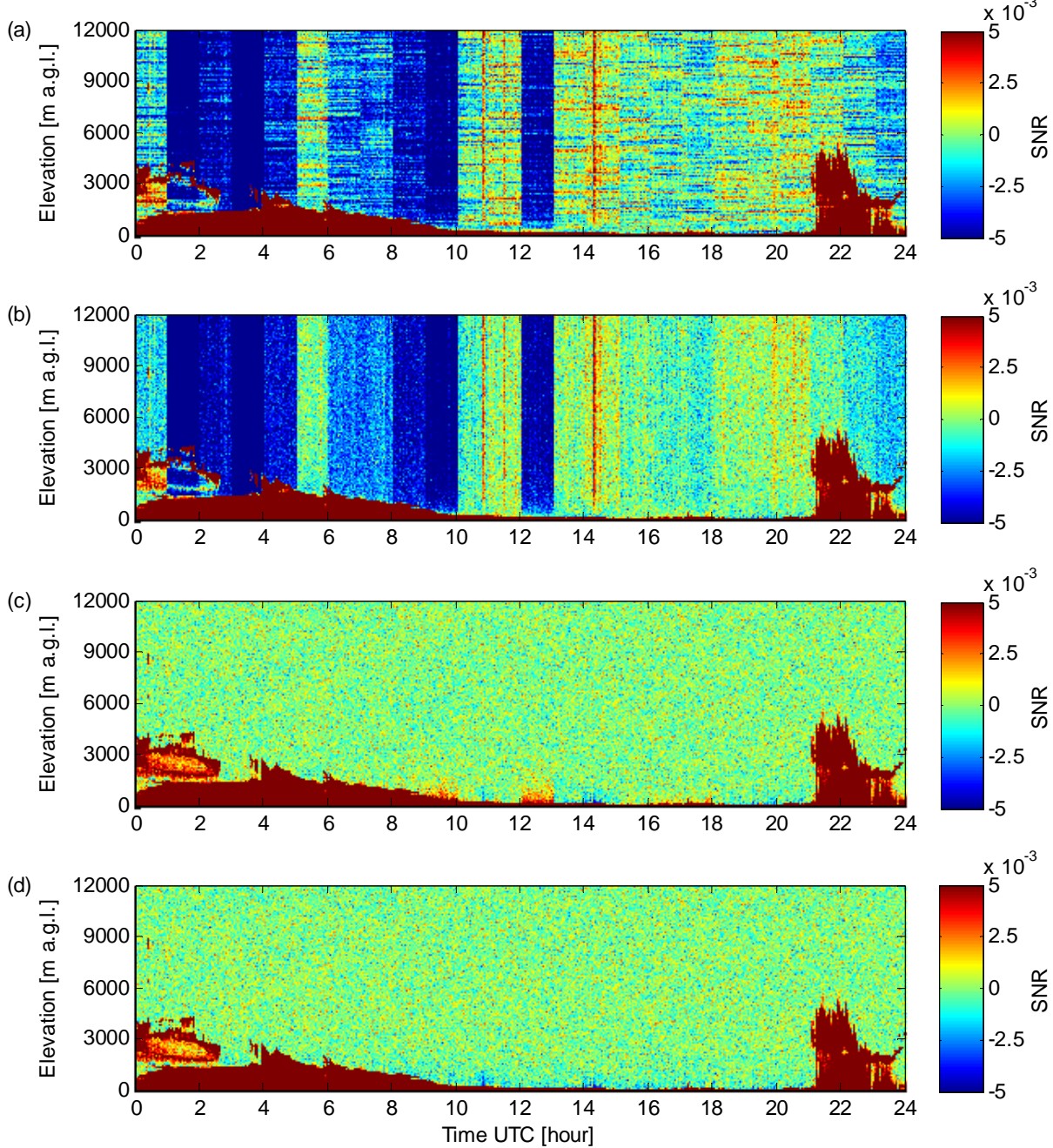

**Figure 6: Data from lidar 146 at Helsinki on 6 May 2018. (a) Time series of** *SNR₀* **profile in vertically-pointing mode. (b) Time series of** *SNR₁* **profile in vertically-pointing mode. (c) Time series of** *SNR₂* **profile in vertically-pointing mode. (d) Time series of** *SNR'₂* **(based always on a linear fit to** $P_{bkg}(z)$**) profile in vertically-pointing mode.**

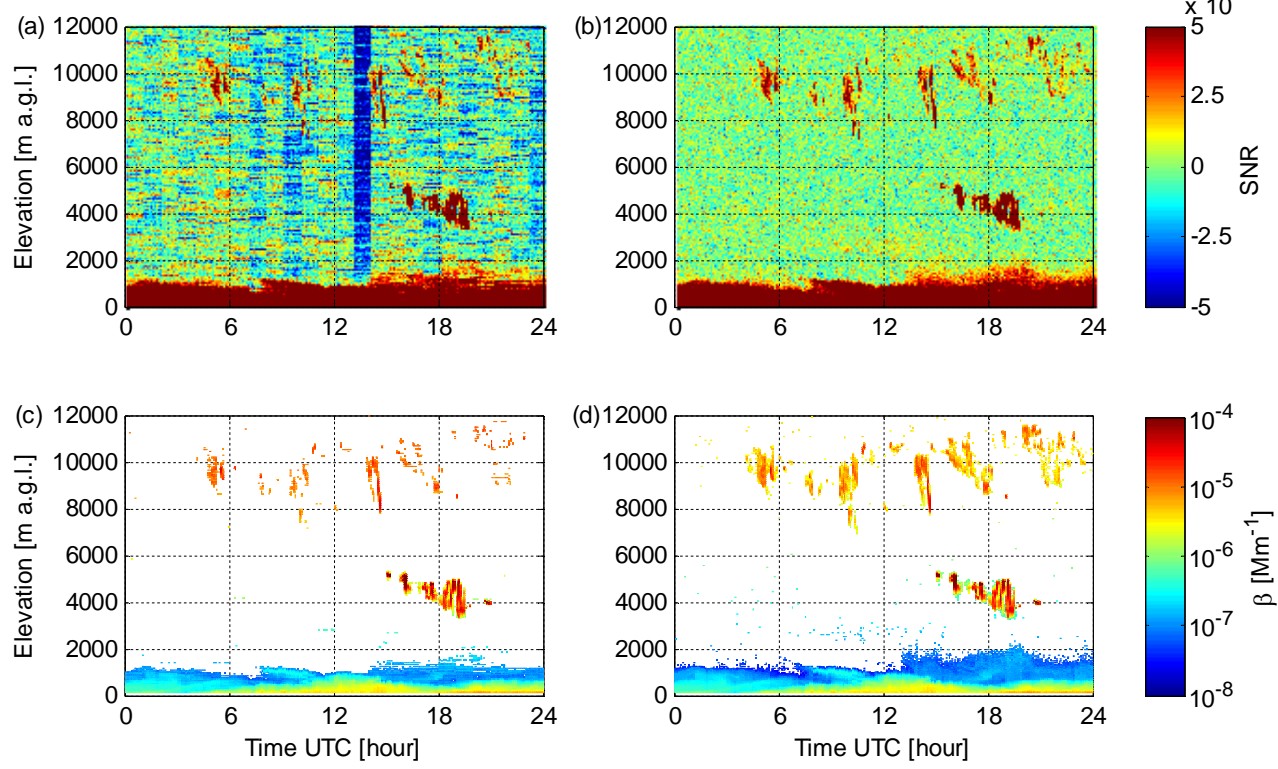

**Figure 7: Data from lidar 146 at Helsinki on 1 May 2018. (a) Time series of $SNR_0$ profile in vertically-pointing mode. (b) Time series of $SNR_2$ profile in vertically-pointing mode. (c) Time series of β obtained with 60s integration time from $SNR_0$. β has been filtered with a threshold of 0.0035 (3σ) applied to $SNR_0$. (d) Time series of β obtained with 60s integration time from $SNR_2$. β has been filtered with a threshold of 0.0012 (3σ) applied to $SNR_2$.**

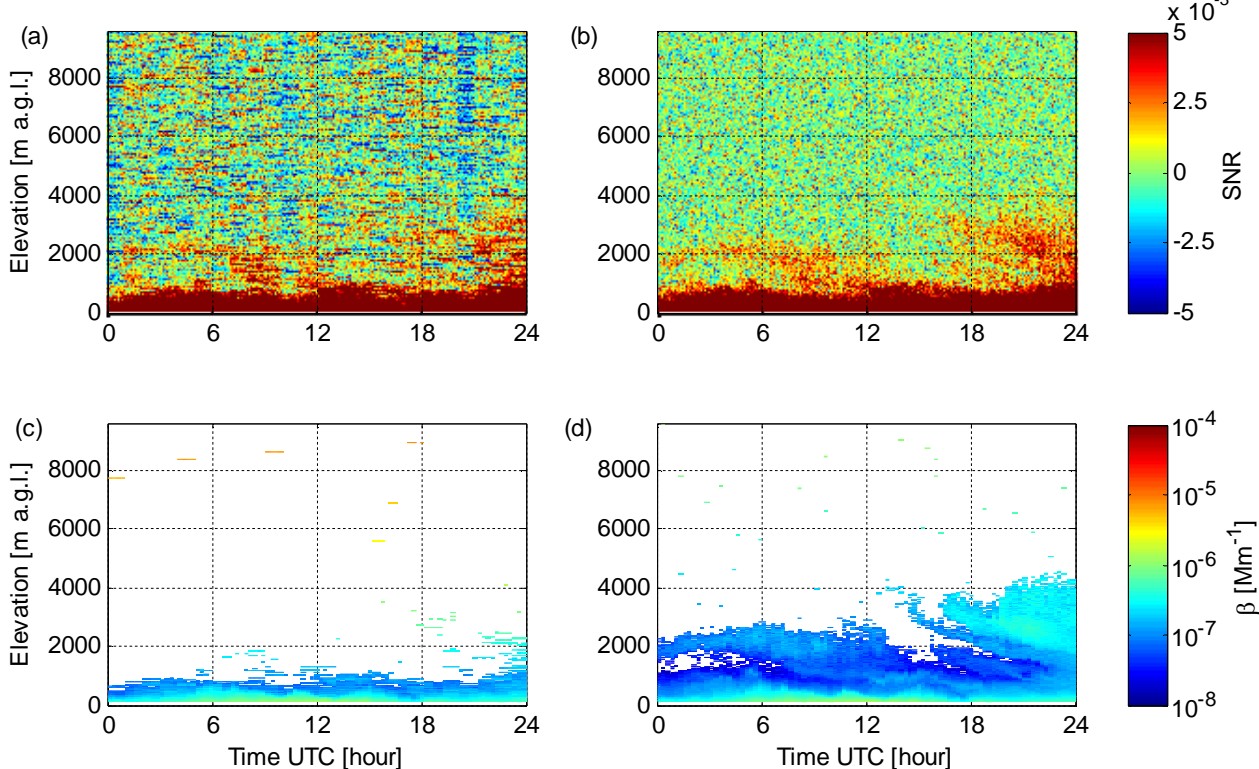

**Figure 8: Data from lidar 53 at Finokalia on 8 July 2014. (a) Time series of $SNR_0$ profile in vertically-pointing mode. (b) Time series of $SNR_2$ profile in vertically-pointing mode. (c) Time series of β obtained with 350s integration time from $SNR_0$. β has been filtered with a threshold of 0.0044 (3σ) applied to $SNR_0$. (d) Time series of β obtained with 350s integration time from $SNR_2$. β has been filtered with a threshold of 0.00059 (3σ) applied to $SNR_2$.**

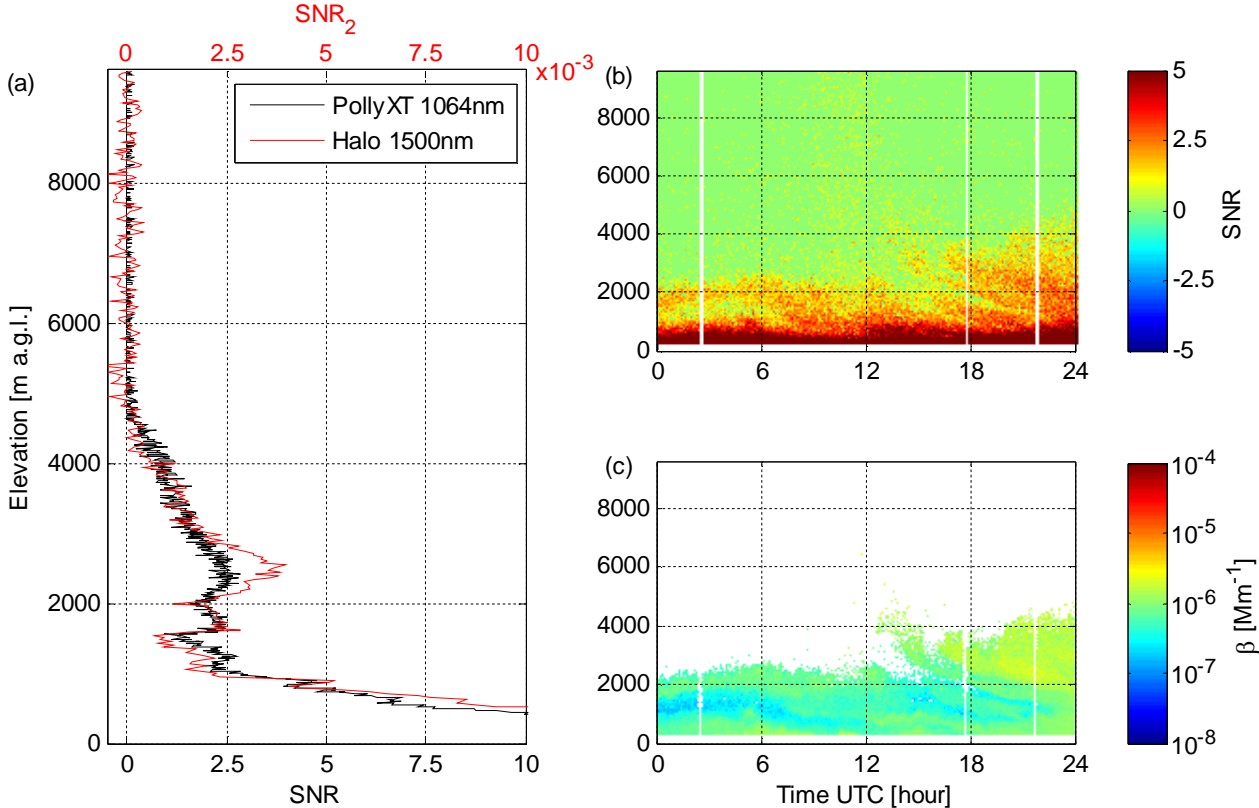

**Figure 9: (a) Vertical profiles of SNR from PollyXT at 1064nm wavelength and SNR$_2$ from lidar 53 at Finokalia on 8 July 2014. Both profiles are obtained at 21 UTC; integration time of lidar 53 profile is 350s and integration time of PollyXT profile is 360s. (b) Time series of PollyXT SNR at 1064nm wavelength with 360s integration time at Finokalia on 8 July 2014. (c) Time series of PollyXT attenuated backscatter at 1064nm wavelength with 360s integration time at Finokalia on 8 July 2014.**