# Peer review of "A novel post-processing algorithm for Halo Doppler lidars"

_Atmospheric Measurement Techniques, 2018_

## Referee Comment (RC1) · Anonymous Referee #1 · 14 Nov 2018

The paper presents a new background correction method for the HALO Photonics Streamline Doppler lidar, but it can also be applied to other Doppler lidar systems. The paper is a well structured and the algorithm description is presented in detail. My overall impression of the paper is good and it should be published in AMT if the authors take into account the following points:

Major issues:

- The method is actually quite powerful and significantly lifts the performance of the Doppler lidar systems. It comes even close to the sensitivity of more powerful Raman lidar systems. In page 8 line 27 such collocated measurements with a Raman lidar are mentioned. It would be nice to a) show attenuated backscatter data of this Raman lidar system together with the improved signals in Fig. 8b. b) show profiles of att.

[Figure]

backscatter of Raman and Doppler lidars to prove that there is no remaining trend with height in the corrected data. Such an analysis would give additional value to the paper and help the reader to perceive the performance of this post-processing algorithm.

- Such work should reflect back on the next versions of the lidar systems and/or the on-board processing software. The algorithm should even be included in the data acquisition itself. Please comment if there are any plans for that.

- Is the software available somewhere? It would be really useful to point to a repository of any kind.

- Page 9 line 17: "With enhanced SNR, the instrumental contribution to radial velocity variance can be estimated with better accuracy". The method discussed in the paper is only applied to the SNR. How does improvement of radial velocity work in this context? Is the improvement maybe based on a better selection of radial velocity values? This is not fully clear, yet very interesting and should be demonstrated in the paper. Please provide plots of enhanced radial velocity together with Fig. 5, 7 or 8.

Minor issues:

- Would temperature stabilization of the Stream Line / Stream Line Pro instruments help to reduce noise?

- The authors present an algorithm that is applied to the evaluated profiles of SNR. Would it make sense to apply it to the raw spectra.

- Concerning the alternation of the emitter of lidar 146: Is it a fault of the individual system or is it found for all systems?

Technical issues:

- Table 1 could be condensed. Stream Line and Stream Line Pro seem to be identical and the only difference to the Stream Line XR seems to be pulse repetition rate and maximum range.

[Figure]

- Figure 2: Using red and black both in (a) and (c) is a bit confusing in this context. Please use alternative colors in (c).

[Figure]

---

## Referee Comment (RC2) · Anonymous Referee #2 · 5 Dec 2018

General:

The authors present an algorithm to improve the accuracy of the instrumental noise of Halo Photonics Doppler lidars. Therefore, longer integration times become possible; enabling to obtain signals down to -32 dB. This is particularly useful under conditions with low aerosol load.

The algorithm combines two background correction methods. The first method corrects for a variable offset in SNR at each range gate, visible as horizontal stripes in the time-height cross-sections of retrieved backscatter. The description of the source of this error in SNR as well as its correction is novel. The second method corrects for an error in scaling of the raw signal, which can result in vertical stripes in the time-height cross-sections of retrieved backscatter. The second correction method has already

been published in AMT (Manninen et al., 2016). The authors show that combining both methods gives added value to the retrieval of atmospheric turbulence properties as well as aerosol observations. Additionally, the "novel post-processing" algorithm could be used in other Doppler lidars and other lidar systems.

Overall, the paper is well written and the description of the algorithm is logical and well-structured. However, the authors should include a more detailed discussion of SNR statistics.

Therefore, the given paper presents a valuable contribution to the field of lidar-based remote sensing. I recommend it for publication in AMT after considering the following comments.

Major comments:

1) Does applying the horizontal stripe correction prior to the Manninen correction influence the performance of the cloud screening, which included in the latter?

2) In Figures 6, 7 and 8 the reader can visually see the improvements in SNR looking at SNR0 to SNR2, but it is essential to quantify the improvement, e.g. with a figure along the line of Figures 8 or 9 in Manninen et al. (2016). Such a statistical analysis of the SNR would present a more detailed picture of the algorithms performance. It would also help to answer comment 1).

3) p.2 l. 18: "as uncertainty . . . wind retrievals. . ." + p.9 l.17-21: "With enhanced SNR,. . ."

On the one hand, the introduction refers not only to turbulence but also to wind retrievals. On the other, the conclusion only states turbulence retrievals. Methods, such as velocity azimuth display utilize the SNR to determine reliable radial velocity measurements. Therefore, the correction algorithm increases the data availability and probably decreases the uncertainty in the retrieval.

The paper would gain more attention if this is aspect is included in the discussion;

either by actually showing that the novel post-processing improves the retrieved winds or by including a paragraph on the expected benefit for wind retrievals.

4) Manninen et al. (2016) give the code as well as an example as supporting information. This paper's algorithm could spread a lot faster, if the authors also include the code in the supporting information or point to a repository, e.g. on github.

Minor comments:

1) Figure 1b shows the temperature dependency of the background profiles. Between 20°C and ca. 27°C the increase seems linear with only a small spread. Above 27°C the increase appears linear with a steeper slope and wider spread of values. Is it possible that part of this change (especially the wider spread) is due to the switch on of the active cooling unit of the Halo Streamline? The cooling would insert vibration and additional electronic noise into the lidar system which could alter the background measurement.

2) Figure 2 and the corresponding discussion on page 5 show that even if the technical specifications of two Doppler lidars are the same (see Table 1), their performance can be different due to instrumental characteristics. This presents an additional finding of this paper which is especially important when operating multiple Doppler lidars. The users should be aware of the possible differences in instruments of the same type. Hence, this finding deserves more attention and should be mentioned in the conclusion.

3) Concerning the high and low mode in the XR lidar: Do the authors know by now, why there are two modes?

Halo lidars can be equipped with a depolarization channel. Is instrument number 146 such an instrument and can the authors exclude that the seemingly random shifts in modes are due to switching between co- and cross-polarization?

4) Could methods similar to the horizontal stripe correction be applied to other lidar

systems such as ceilometers?

5) Some readers could be interested in using Doppler lidars semi-operationally, e.g. for a seasonal campaign or even 24/7. Which parts of the algorithm are applicable operationally? How long does the data set have to be in order to successfully apply the post-processing? What about longer data gaps?

The publication would benefit from a remark to operational applicability.

6) The introduction starts by mentioning "turbulent mixing" and "mixing layer height", but the paper presents no estimate of the MLH in one of the Figures 4 through 8. Including the MLH as a line plot in Figure 4 or elsewhere would round up the discussion that started in the introduction.

7) p.5 l. 11: "t"elescope.

8) Table 1 seems to long and could be shortened by mentioning redundant information only once

---

## Author Comment (AC1) · 16 Jan 2019

Response to Anonymous Referee 1

Referee comments are indicated in *italic*, followed by our reply.

*The paper presents a new background correction method for the HALO Photonics Streamline Doppler lidar, but it can also be applied to other Doppler lidar systems. The paper is a well structured and the algorithm description is presented in detail. My overall impression of the paper is good and it should be published in AMT if the authors take into account the following points:*

We would like to thank the referee for the valuable comments, which have improved the manuscript.

*Major issues:*

*- The method is actually quite powerful and significantly lifts the performance of the Doppler lidar systems. It comes even close to the sensitivity of more powerful Raman lidar systems. In page 8 line 27 such collocated measurements with a Raman lidar are mentioned. It would be nice to a) show attenuated backscatter data of this Raman lidar system together with the improved signals in Fig. 8b. b) show profiles of att. backscatter of Raman and Doppler lidars to prove that there is no remaining trend with height in the corrected data. Such an analysis would give additional value to the paper and help the reader to perceive the performance of this post-processing algorithm.*

Thank you for the suggestion. We have added Raman lidar SNR and attenuated backscatter profiles as a new Fig. 9. Fig 9 includes also a comparison of Raman and Doppler lidar SNR profiles at 21 UTC. We have added a short description of the Raman system in Section 2 and modified text on page 8, line 26-27 as:

"These aerosol layers were also observed with a co-located multi-wavelength Raman lidar PollyXT (Baars et al., 2016; Engelman et al., 2016). A comparison of lidar 53 and the Raman lidar measurements at 1064 nm wavelength are presented in Fig. 9. Considering the wavelength difference, the agreement between the two systems is reasonably good."

[Figure]

**Figure 9: (a) Vertical profiles of SNR from PollyXT at 1064nm wavelength and SNR$_2$ from lidar 53 at Finokalia on 8 July 2014. Both profiles are obtained at 21 UTC; integration time of lidar 53 profile is 350s and integration time of PollyXT profile is 360s. (b) Time series of PollyXT SNR at 1064nm wavelength with 360s integration time at Finokalia on 8 July 2014. (c) Time series of PollyXT attenuated backscatter at 1064nm wavelength with 360s integration time at Finokalia on 8 July 2014.**

*- Such work should reflect back on the next versions of the lidar systems and/or the on-board processing software. The algorithm should even be included in the data acquisition itself. Please comment if there are any plans for that.*

To our knowledge there are no plans to incorporate this algorithm in the on-board software at the moment. In our opinion this is not a severe limitation, as most (if not all) lidar systems require some post-processing for optimal data quality.

*- Is the software available somewhere? It would be really useful to point to a repository of any kind.*

A Matlab implementation of the algorithm presented here is available at https://github.com/manninenaj/HALO_lidar_toolbox. We have added a note on this in Conclusion and a reference to the toolbox in the references.

*- Page 9 line 17: "With enhanced SNR, the instrumental contribution to radial velocity variance can be estimated with better accuracy". The method discussed in the paper is only applied to the SNR. How does improvement of radial velocity work in this context? Is the improvement maybe based on a better selection of radial velocity values? This is not fully clear, yet very interesting and should be demonstrated in the paper. Please provide plots of enhanced radial velocity together with Fig. 5, 7 or 8.*

This method does not affect radial velocity values from the Doppler lidar, but only SNR.

As discussed in the manuscript on page 3, lines 17-20, instrumental uncertainty in radial velocity is a function of SNR (see also Fig. 3b). Variance calculated from a set of observed radial velocities will have contributions from both instrumental uncertainty and atmospheric turbulence. By improving the accuracy of SNR, this method enables more accurate determination of the instrumental contribution, which can be substantial under low signal conditions. This will result in more accurate retrieval of turbulent parameters when instrumental noise contribution is subtracted from the observed radial velocity variance.

We have tried to clarify this sentence as
"With enhanced SNR, the instrumental noise contribution to radial velocity variance can be estimated with better accuracy, which will improve the quality of turbulent parameter retrievals."

*Minor issues:*

*- Would temperature stabilization of the Stream Line / Stream Line Pro instruments help to reduce noise?*

Temperature stabilisation would help to ensure that the shape of Pamp is known well (c.f. Fig. 2b). However, temperature dependency of Pamp is a relatively small source of noise in the SNR, in the order of magnitude $10^{-5}$ x (SNR+1). We have added a note to this effect on page 5, line 23:

"For optimal data quality, additional temperature stabilisation could be applied to ensure that $P_{amp}$ is always in the well-characterised temperature range."

*- The authors present an algorithm that is applied to the evaluated profiles of SNR. Would it make sense to apply it to the raw spectra.*

If SNR is calculated from the raw spectra, this algorithm will be needed to determine the actual noise floor. We have added a note to this effect in conclusions on page 9, line 16.

*- Concerning the alternation of the emitter of lidar 146: Is it a fault of the individual system or is it found for all systems?*

We have observed this behaviour on the two systems with the new, more sensitive amplifier that we have access to at the moment. We consider it likely that this alternation will be present in all systems with the XR-amplifier. This is easy to check from the saved background files by the lidar operator, though.

*Technical issues:*

*- Table 1 could be condensed. Stream Line and Stream Line Pro seem to be identical and the only difference to the Stream Line XR seems to be pulse repetition rate and maximum range.*

Table 1 has been condensed:

**Table 1 Specifications for Halo Doppler lidars utilised in this study.**

| | |
|---|---|
| Lidar number and version | 46, Stream Line |
| | 53, Stream Line Pro |
| | 146, Stream Line XR |
| Wavelength | 1.5 μm |
| Pulse repetition rate | 15 kHz (46 and 53) or |
| | 10 kHz (146) |
| Nyquist velocity | 20 m s$^{-1}$ |
| Sampling frequency | 50 MHz |
| Velocity resolution | 0.038 m s$^{-1}$ |
| Points per range gate | 10 |
| Range resolution | 30 m |
| Maximum range | 9600 m (46 and 53) or |
| | 12000 m (146) |
| Pulse duration | 0.2 μs |
| Lens diameter | 8 cm |
| Lens divergence | 33 μrad |
| Telescope | monostatic optic-fibre |
| | coupled |

*- Figure 2: Using red and black both in (a) and (c) is a bit confusing in this context. Please use alternative colors in (c).*

We have changed the colors in Fig. 2c to blue and black:

[Figure]

**Figure 2: (a) Lidar 46** $P_{bkg,res}$ **averaged from 20 August 2016 to 14 June 2017 (7193 background checks). Lidar 53** $P_{bkg,res}$ **averaged from 1 January 2014 to 30 November 2015 (16802 background checks).** $P_{amp}$ **is plotted for both systems. (b) First 100 range gates of lidar 46** $P_{amp}$ **calculated for different ranges of T. (c) Lidar 146** $P_{bkg,res}$ **averaged from 12 January 2018 to 31 May 2018.** $P_{bkg,res}$ **is averaged separately for high** $P_{bkg}$ **mode (1375 background checks) and for low** $P_{bkg}$ **mode (1623 background checks).** $P_{amp}$ **is plotted for both modes.**

---

## Author Comment (AC2) · 16 Jan 2019

Response to Anonymous Referee 2

Referee comments are indicated in *italic*, followed by our reply.

*General:*
*The authors present an algorithm to improve the accuracy of the instrumental noise of Halo Photonics Doppler lidars. Therefore, longer integration times become possible; enabling to obtain signals down to -32 dB. This is particularly useful under conditions with low aerosol load.*
*The algorithm combines two background correction methods. The first method corrects for a variable offset in SNR at each range gate, visible as horizontal stripes in the time-height cross-sections of retrieved backscatter. The description of the source of this error in SNR as well as its correction is novel. The second method corrects for an error in scaling of the raw signal, which can result in vertical stripes in the time-height cross-sections of retrieved backscatter. The second correction method has already been published in AMT (Manninen et al., 2016). The authors show that combining both methods gives added value to the retrieval of atmospheric turbulence properties as well as aerosol observations. Additionally, the "novel post-processing" algorithm could be used in other Doppler lidars and other lidar systems.*
*Overall, the paper is well written and the description of the algorithm is logical and well-structured. However, the authors should include a more detailed discussion of SNR statistics. Therefore, the given paper presents a valuable contribution to the field of lidar-based remote sensing. I recommend it for publication in AMT after considering the following comments.*

We would like to thank the referee for the valuable comments, which have improved the manuscript.

*Major comments:*

*1) Does applying the horizontal stripe correction prior to the Manninen correction influence the performance of the cloud screening, which included in the latter?*

Applying the horizontal stripe correction makes it easier to perform cloud screening, as it is easier to discern thin clouds and aerosol layers. We have added a sentence on page 6, line 3:

"Note that typically cloud and aerosol signal is easier to discern in $SNR_1$ than in $SNR_0$ and thus cloud screening is applied after Equation 5."

*2) In Figures 6, 7 and 8 the reader can visually see the improvements in SNR looking at SNR0 to SNR2, but it is essential to quantify the improvement, e.g. with a figure along the line of Figures 8 or 9 in Manninen et al. (2016). Such a statistical analysis of the SNR would present a more detailed picture of the algorithms performance. It would also help to answer comment 1).*

In our opinion the major advantage of this algorithm is that it enables averaging SNR to utilise very weak signals, which is illustrated in Fig. 4d. However, to give more detailed picture of the algorithm's performance we have added histograms of $SNR_0$ and $SNR_2$ in cloud and aerosol free regime for the four cases studies as an Appendix to the manuscript. The new Fig. A1 is below.

[Figure]

**Figure A1. Histograms of SNR$_0$ and SNR$_2$ in cloud and aerosol free regime for the four case studies considered in Section 4. For each case, mean [standard deviation] and median [25th, 75th percentile] of SNR$_0$ and SNR$_2$ are included. (a) Welgegund on 6 September 2016, 00-24 h UTC, 4800-9000 m a.g.l.. (b) Kumpula on 1 May 2018, 02-24 h UTC, 6000-12000 m a.g.l.. (c) Kumpula on 6 May 2018, 00-12 h UTC, 4000-7000 m a.g.l.. (d) Finokalia on 8 July 2014, 00-24 h UTC, 5000-96000 m a.g.l..**

*3) p.2 l. 18: "as uncertainty : : : wind retrievals: : :" + p.9 l.17-21: "With enhanced SNR,: : :"*
*On the one hand, the introduction refers not only to turbulence but also to wind retrievals.*
*On the other, the conclusion only states turbulence retrievals. Methods, such as velocity azimuth display utilize the SNR to determine reliable radial velocity measurements. Therefore, the correction algorithm increases the data availability and probably decreases the uncertainty in the retrieval.*
*The paper would gain more attention if this is aspect is included in the discussion; either by actually showing that the novel post-processing improves the retrieved winds or by including a paragraph on the expected benefit for wind retrievals.*

Thank you for bringing this up. Wind retrieval methods should be able to discard outliers (e.g. Päschke et al., 2015), so we do not expect that the retrieved wind speed and direction would be affected. However, as a lower SNR-threshold can be applied after the new post-processing, the data availability will increase. How large this effect is depends on atmospheric conditions. We have added following discussion in Section 4.1, where also vertical winds are discussed:

"Lower noise floor enables also wind retrievals with a lower SNR-threshold, which increases the data availability. The effect on data availability depends on atmospheric conditions, though. In this

case for instance (Welgegund, 6 September 2016) a 75° elevation angle velocity azimuth display (VAD) scan was utilised for horizontal wind retrieval every 15 minutes. With the new post-processing algorithm the SNR-threshold for wind retrieval could be decreased from 0.0045 to 0.0032. This decrease in SNR-threshold enabled wind retrieval for 2-13 range gates more from each VAD scan; on average, winds could be determined from 7.5 additional range gates per VAD scan. That is, vertical coverage of wind retrievals increased on average by 200 m with the new post-processing.

Wind retrievals at lower SNR will have higher uncertainty due to higher instrumental noise in radial velocity measurement, yet enhanced SNR will enable more accurate determination of the instrumental uncertainty in wind retrievals. However, as a major fraction of the uncertainty in retrieved winds arises in atmospheric turbulence (Newsom et al., 2017) the more accurate SNR will have only a limited effect on the overall uncertainty in the wind retrieval. Therefore, the uncertainty in each wind retrieval should be evaluated e.g. with the methodology of Newsom et al. (2017) before the wind retrievals are disseminated."

We have added a note on improved data coverage for horizontal wind also in the Conclusion.

*4) Manninen et al. (2016) give the code as well as an example as supporting information. This paper's algorithm could spread a lot faster, if the authors also include the code in the supporting information or point to a repository, e.g. on github.*

A Matlab implementation of the algorithm presented here is available at https://github.com/manninenaj/HALO_lidar_toolbox. We have added a note on this in Conclusion and a reference to the toolbox in the references.

*Minor comments:*

*1) Figure 1b shows the temperature dependency of the background profiles. Between 20_C and ca. 27_C the increase seems linear with only a small spread. Above 27_C the increase appears linear with a steeper slope and wider spread of values. Is it possible that part of this change (especially the wider spread) is due to the switch on of the active cooling unit of the Halo Streamline? The cooling would insert vibration and additional electronic noise into the lidar system which could alter the background measurement.*

It is quite possible that this change coincides with the switching on of the fans. However, we did not find temperature dependency in the actual measurement noise (measured as the variance of SNR for cloud and aerosol free range gates). The only temperature effect we found is in $P_{amp}$ (Fig. 2b), which becomes relevant only when integration time is several minutes or longer.

*2) Figure 2 and the corresponding discussion on page 5 show that even if the technical specifications of two Doppler lidars are the same (see Table 1), their performance can be different due to instrumental characteristics. This presents an additional finding of this paper which is especially important when operating multiple Doppler lidars. The users should be aware of the possible differences in instruments of the same type. Hence, this finding deserves more attention and should be mentioned in the conclusion.*

Indeed, each lidar system needs to be individually characterised by the lidar operator. We have added following paragraph in conclusion:

"Our analysis shows that even if the technical specifications of two Doppler lidar systems are identical, their instrumental noise characteristics can be quite different (Fig. 2). Therefore, the lidar operator should inspect each system individually to ensure highest data quality. Note that this algorithm or similar processing is needed to define the instrumental noise level even if raw spectra are utilised instead of the processed in data. The algorithm presented here can be applied in semi-operational use as long as at least 300 background checks (acquired in two weeks of measurements with typical configuration) are available for characterising the amplifier response to the transmitted pulse. A Matlab implementation of this algorithm is available through Github (Manninen, 2019)."

Reference:

Manninen, A: HALO lidar toolbox, GitHub, https://github.com/manninenaj/HALO_lidar_toolbox, 2019.

*3) Concerning the high and low mode in the XR lidar: Do the authors know by now, why there are two modes? Halo lidars can be equipped with a depolarization channel. Is instrument number 146 such an instrument and can the authors exclude that the seemingly random shifts in modes are due to switching between co- and cross-polarization?*

All instruments utilised in this study are equipped with a depolarization channel. However, the software is configured to do background check always in the co-polar mode. Our understanding is that this is a feature of the amplifier. However, whether these two modes are present in a system is easy to check by the lidar operator from the background files. Note also that some systems are configured with 10kHz pulse repetition frequency, but use the non-XR amplifier.

*4) Could methods similar to the horizontal stripe correction be applied to other lidar systems such as ceilometers?*

This kind of method can be applied to any lidar system that defines noise level from a finite duration background check (c.f. page 9, line 8-10). However, ceilometers do not routinely carry out such background measurement – in ceilometers the most important source of noise is solar background, which requires its own processing.

*5) Some readers could be interested in using Doppler lidars semi-operationally, e.g. for a seasonal campaign or even 24/7. Which parts of the algorithm are applicable operationally? How long does the data set have to be in order to successfully apply the post-processing? What about longer data gaps? The publication would benefit from a remark to operational applicability.*

The algorithm is not computationally heavy and can be utilised semi-operationally with time delay of a few minutes from measurement to post-processed data. As pointed out on page 5, line 19-20, approximately 300 background checks are needed to obtain a reliable estimate of $P_{amp}$. This corresponds to two weeks of measurements with hourly background checks. Such short record does not facilitate accounting for temperature dependency of $P_{amp}$, but that is a minor effect compared to other sources of uncertainty in the noise level.

As long as a continuous record of background checks are available, data gaps are not an issue for this post-processing algorithm.

*6) The introduction starts by mentioning "turbulent mixing" and "mixing layer height", but the paper presents no estimate of the MLH in one of the Figures 4 through 8. Including the MLH as a line plot in Figure 4 or elsewhere would round up the discussion that started in the introduction.*

We have added MLH as a line in Fig. 5.

*7) p.5 l. 11: "t"elescope.*

Corrected on page 3, line 11.

*8) Table 1 seems to long and could be shortened by mentioning redundant information only once*

Table 1 has been condensed:

**Table 1 Specifications for Halo Doppler lidars utilised in this study.**

| | |
|---|---|
| Lidar number and version | 46, Stream Line |
| | 53, Stream Line Pro |
| | 146, Stream Line XR |
| Wavelength | 1.5 μm |
| Pulse repetition rate | 15 kHz (46 and 53) or |
| | 10 kHz (146) |
| Nyquist velocity | 20 m s$^{-1}$ |
| Sampling frequency | 50 MHz |
| Velocity resolution | 0.038 m s$^{-1}$ |
| Points per range gate | 10 |
| Range resolution | 30 m |
| Maximum range | 9600 m (46 and 53) or |
| | 12000 m (146) |
| Pulse duration | 0.2 μs |
| Lens diameter | 8 cm |
| Lens divergence | 33 μrad |
| Telescope | monostatic optic-fibre |
| | coupled |